# Spirocyclic Nitroxides as Versatile Tools in Modern Natural Sciences: From Synthesis to Applications. Part I. Old and New Synthetic Approaches to Spirocyclic Nitroxyl Radicals

**DOI:** 10.3390/molecules26030677

**Published:** 2021-01-28

**Authors:** Elena V. Zaytseva, Dmitrii G. Mazhukin

**Affiliations:** Novosibirsk Institute of Organic Chemistry, Siberian Branch of Russian Academy of Sciences (SB RAS), Academician Lavrentiev Ave. 9, 630090 Novosibirsk, Russia; elena@nioch.nsc.ru

**Keywords:** spirocyclic nitroxides, recyclization, condensation, 1,3-dipolar cycloaddition, TEMPO, PROXYL, 3-imidazoline nitroxides, DOXYL, bis-nitroxides, molecular structure, EPR

## Abstract

Spirocyclic nitroxyl radicals (SNRs) are stable paramagnetics bearing spiro-junction at α-, β-, or γ-carbon atom of the nitroxide fragment, which is part of the heterocyclic system. Despite the fact that the first representatives of SNRs were obtained about 50 years ago, the methodology of their synthesis and their usage in chemistry and biochemical applications have begun to develop rapidly only in the last two decades. Due to the presence of spiro-function in the SNRs molecules, the latter have increased stability to various reducing agents (including biogenic ones), while the structures of the biradicals (SNBRs) comprises a rigid spiro-fused core that fixes mutual position and orientation of nitroxide moieties that favors their use in dynamic nuclear polarization (DNP) experiments. This first review on SNRs will give a glance at various strategies for the synthesis of spiro-substituted, mono-, and bis-nitroxides on the base of six-membered (piperidine, 1,2,3,4-tetrahydroquinoline, 9,9′(10*H*,10*H*′)-spirobiacridine, piperazine, and morpholine) or five-membered (2,5-dihydro-1*H*-pyrrole, pyrrolidine, 2,5-dihydro-1*H*-imidazole, 4,5-dihydro-1*H*-imidazole, imidazolidine, and oxazolidine) heterocyclic cores.

## 1. Introduction

Stable nitroxyl radicals (NRs), first obtained more than 150 years ago (Figure 1, **1a**, R_1,__2_ = SO_3_K, Frémy’s salt), became widely known with the development of electron paramagnetic resonance (EPR) spectroscopy methods and thanks to the discovery in 1959 of extremely stable cyclic derivatives (Figure 1, **1b**, R_1–4_ = Me) of piperidine series (TEMPO) [1]. Over the past 60 years, NRs have become an integral part of scientific research related to the development of advanced methods for the construction of new materials for chemistry, biochemical studies, medical applications, and energy storage needs.

Only in the last decade, many monographs, book chapters, and reviews were published regarding the physicochemical features of these radicals, methods of their synthesis, and their various applications [2,3,4,5] in synthetic organic chemistry (as oxidants) [6], the creation of functional materials [7,8,9,10], and polymer chemistry [11], including their role as agents in nitroxide-mediated radical polymerization [12,13,14], as unique molecular spin probes [15] and indispensable spin labels [16,17], and antioxidants and potential therapeutic agents for biological research and medicine [18,19,20].

At the same time, these monographs and reviews only occasionally or incompletely mention the so-called spirocyclic nitroxyl radicals (SNRs), which include a wide range of nitroxides containing spirocyclic moiety(-ies) at the α-, β-, or γ-carbon atoms near a paramagnetic center (Figure 1, structures **2a–c**).

SNRs, especially those belonging to α-SNRs (Figure 1, **2a**) or containing several spiro moieties—in comparison with traditional NRs (Figure 1, **1a**, R_1,__2_ = *tert*-Alk; R_1_ = *t*-Bu; R_2_ = Ar; R_1,__2_ = Ar) or with those in Figure 1, **1b**, possessing a tetra-alkyl environment near the paramagnetic center—have a number of definite advantages, e.g.,

(a) they have a much lower rate of radical center reduction by biogenic reducing agents owing to a decrease in the steric accessibility of the N–O group;

(b) due to the hindered rotation of the spiro moiety, they have spin-echo dephasing time (*T_m_* ) long enough up to ~125 K to allow for double electron-electron resonance (DEER) measurements of interspin distances in the liquid-nitrogen temperature range; 

(c) they offer an opportunity for fine-tuning the changes in the properties of the radical (hydrophilicity/hydrophobicity) or (open/closed) conformation (Figure 2) via the introduction of proper substituents or heteroatoms at different positions of the spirocycle. Furthermore, the functional group in the spiro moiety could be used as a spin label in molecular biology or as a spin-labeled chelate-forming reagent in analytical chemistry;

(d) the spiro functions can be employed as protective groups; under certain conditions, they can be destroyed using an acidic medium or enzymatically, leading to the release of active proton-containing groups;

(e) the spiro substituent can serve as a rigid linker to create di- or polyradical molecules, whose magnitude of the exchange interaction between paramagnetic centers can be managed by changing the size or geometry of the linker;

(f) the introduction of some chirality into the spiro function can predetermine specific physicochemical properties of the SNR and makes it possible to use the radical as a chiral auxiliary in stereoselective processes.

Here, it is possible to further enumerate the benefits of SNRs, but it is better to briefly refer to the current state of affairs in terms of their application in various fields of chemistry, biology, and physics.

During the last several decades, numerous research groups from the USA, France, Japan, Germany, and Russia have shown that SNRs are applicable as:

(a) sterically shielded spin labels and spin probes with high resistance to biological reductants having long spin-spin relaxation times [21,22,23,24,25,26,27,28,29,30,31,32,33,34,35,36,37,38,39,40,41];

(b) paramagnetic agents for dynamic nuclear polarization (DNP) both in solutions and as solids [42,43,44,45,46,47,48,49,50,51,52,53,54,55,56,57,58,59,60,61,62,63,64,65];

(c) catalysts for living radical polymerization [66,67,68,69,70,71,72,73,74,75];

(d) starting compounds for the synthesis of oxoammonium salts, which are used as oxidants in organic chemistry [76] and for the creation of organic radical batteries [77];

(e) building blocks for magnetic materials [78,79,80,81];

(f) organic radical contrast agents (ORCAs) for magnetic resonance imaging (MRI) [82,83,84,85,86,87,88,89];

(g) as participants in host–guest interactions in supramolecular chemistry for the creation of nanomachines and polyradical ensembles [90,91,92,93,94,95,96,97,98,99];

(h) as spin probes for studying liquid crystalline media [100,101,102,103]; etc. 

Usually, the preparation of SNRs requires an appliance of nonstandard synthetic strategies that are different from those used for non-spirocyclic analogs. However, no special review dedicated to SNRs synthesis has been published yet. In this regard, the purpose of this survey is to systematize and summarize basic synthetic ways to spirocyclic mono- and bis-nitroxides containing six-membered (piperidine, 1,2,3,4-tetrahydroquinoline, 9,9′(10*H*,10*H’*)-spirobiacridine, piperazine, and morpholine) and five-membered (2,5-dihydro-1*H*-pyrrole, pyrrolidine, 2,5-dihydro-1*H*-imidazole, 4,5-dihydro-1*H*-imidazole, imidazolidine, and oxazolidine) nitrogenous heterocyclic cores.

## 2. Piperidine Nitroxide Radicals (TEMPO Type)

The first SNRs were obtained on the basis of six-membered nitrogen heterocycles. In addition, ring contraction of piperidine NRs is one of the main methods for the synthesis of five-membered nitroxides of the pyrrole series; accordingly, the presentation of data in this review conforms to the principle “from big to small.”

Piperidine-type SNRs (Figure 3, **3**) have been obtained via oxidation of the corresponding sterically hindered amines **4** with hydrogen peroxide or its inorganic and organic analogs (for example, with potassium peroxymonosulfate (OXONE) or *meta*-chloroperoxybenzoic acid [*m*-CPBA]) [104,105,106]. Given that the oxidation step allows us to synthesize SNRs with high yields, the main problem is usually the synthesis of a precursor, a respective amine.

In this regard, the interaction of acetonine (i.e., 2,2,4,6,6-pentamethyl-1,2,5,6- tetrahydropyrimidine **5**, which can be easily prepared from acetone) with different ketones, including cyclic ones, has been studied extensively. A number of catalysts, such as methylammonium chloride, acetic acid, and *para*-toluenesulfonic acid (PTSA), have been tested for this reaction; however, the best result has been obtained for anhydrous ammonium chloride. In addition, the formation of 2,2,6,6-tetramethylpiperidone **6** the product of the interaction between acetonine and acetone, is always observed as a side reaction in this process (Scheme 1). The proportion of ketones **6–8** is determined by the gas chromatography–mass spectrometry (GS–MS) analysis (Table 1). The total isolated yield of mono-**7** and dispirocyclic **8** amounts to 60% when cyclohexanone is used as a ketone. When 2-alkyl-substituted cyclohexanones, including bicyclic ones, (entries 3 and 6–9 in Table 1) are employed in the reaction, dispirocyclic products **8** are not registered in the resultant mixture [76].

A suggested mechanism behind the formation of piperidone **7** is shown in Scheme 2. An initial nucleophilic attack of the tautomeric form of acetonine **9** on the protonated carbonyl compound leads to an intermediate, **10**, which undergoes heterocyclic C–N bond cleavage thereby turning into acyclic compound **11**. Intramolecular cyclization of its tautomer **12** generates piperidine **13**, which is followed by hydrolysis yielding the final product, spirocyclic amine **7**.

Bobbitt et al. synthesized chiral piperidine-type SNRs to obtain the corresponding oxoammonium salts on their basis. The interaction of acetonine **5** with dihydrocarvone (Table 1, entry 6) should give rise to optically active piperidone **14**. Nonetheless, experimentally obtained product **14** is a mixture of isomers. It should be noted that racemization at the C-2 position of the carbocycle takes place during the reaction; meanwhile, the (*R*) configuration of the asymmetric center at the C-5 position of the starting ketone remains the same for all isomers. Moreover, a new asymmetric center at position C-1 is formed. Finally, four diastereomers are obtained with the ratio 49:26:14:11. Spatial structures of the stereoisomers have been established by nuclear magnetic resonance (NMR) spectroscopy. Hydrogenation of isomeric mixture **14** by means of Pt/H_2_ leads to compounds **15**, which are next oxidized by *m*-CPBA to the corresponding radicals **16**. Reductive amination of the keto group in SNR **16** followed by acylation of the intermediate with acetic anhydride leads with a high yield to amide **17**, possessing a new chiral center of an undetermined configuration (Scheme 3) [76].

The synthesis of SNRs **18a–c**, **19b–c**, and **20**, which have been obtained as catalysts for living radical polymerization [67,69,71], can be presented here as an example of the application of the above-mentioned synthetic strategy. It is worth noting that the reaction of the acetonine with cyclic ketones may be the weak link in the whole synthetic scheme. Thus, monospiroamines **21a–c** are obtained as products of the interaction of **5** with cyclopentanone, cyclohexanone, and cycloheptanone, respectively, in the presence of NH_4_Cl or NH_4_Br with rather low yields (from 6% to 14%). Those authors have stated that desired spiro compounds **21a–c** are isolated chromatographically or by vacuum distillation from a complicated reaction mixture. The yields of dispiroamines **22b,c**, which are synthesized under similar experimental conditions, are also low (24% for cyclohexane derivative **22b** and only 1.9% for cycloheptane derivative **22c**). Those authors explain the very low yield of the latter by steric encumbrances preventing its formation [71]. The next step, reduction of ketones **21a–c** and **22b,c** by sodium borohydride to respective amino alcohols **23a–c** and **24b,c**, as a rule, does not pose difficulties. Final oxidation of **23a–c** and **24b,c** by *m*-CPBA or hydrogen peroxide in the presence of disodium salt of ethylenediaminetetraacetic acid (EDTA-Na_2_) and Na_2_WO_4_ gives target SNRs **18a–c** and **19b,c**. A similar oxidation step for **22b** has a yield of only 29%. Such a yield ratio (~3/1) difference between the oxidation of amino alcohol **24b** and the oxidation of aminoketone **22b** is related to their different solubility in ethanol, where the reaction is carried out (Scheme 4).

Spin labeled rotaxanes **25** and **26** are prepared on the basis of an amino derivative of SNR **20** [92]. First, the reaction of SNR **20** with propargylamine, followed by the reduction of the imine, produces radical **27** with a terminal acetylene group. After that, in the key reaction for the formation of [2]rotaxane, radical **27** enters into a “click” reaction with a diazido derivative of 1,8-dialkoxynaphthalene (DAN), **28**, with simultaneous addition of a cyclobis(paraquat-*p*-phenylene) (CBPQT) macrocycle based on paraquat and a catalyst, a copper(I) salt (Scheme 5). A similar rotaxane, **26**, was obtained using a diazido derivative of tetrathiofulvalene (TTF), **29** (Scheme 6). Ideally, cyclobis(paraquat-*p*-phenylene) (CBPQT^4+^)(PF_6_^−^)_4_ having a wheel structure should play the role of a “shuttle” moving due to the redox process in the rotaxane molecule between the “stations” (the corresponding units, DAN (magenta) and TTF (green)), while bulky SNR residues serve as terminal stoppers and EPR-sensitive sensors.

In continuation of this work, Lucarini et al. have managed to implement the following idea: a multifunctional rotaxane **30** is created with the CBPQT shuttle, i.e., a spin (TEMPO)-labeled radical that can move from the TTF core to DAN core, where at the terminus of a molecule, a second label (an SNR stopper) is situated (Figure 4). Single-electron oxidation of the TTF unit results in a significant through-space magnetic interaction between different radical units. In some cases, rotaxanation has proven to have dramatic effects on such magnetic interactions [95].

SNRs **31** and **32** are obtained initially as spin probes from acetonine **5** and 4-hydroxycyclohexanone in two to three steps. The yield of intermediate amine **33** is only 19%. Oxidation of **33** by *m*-CPBA gives the corresponding radical **31**, which is reduced quantitatively by sodium borohydride (without affecting the radical center) to SNR **32** with three hydroxy groups [28]. Based on nitroxide **31**, through its conversion into amino derivative **34** with subsequent transformation into *N,N**′*-disubstituted urea, a polarizing agent for effective DNP of biomolecules is synthesized—biradical **bcTol** (**35**) with high water solubility [58]. 

When methylamine is used in a similar procedure instead of NH_4_OAc, and carbonyldiimidazole is replaced by bis(trichloromethyl)carbonate (BTC), a dimethyl analog of **35**, R*NMe-CO-NMeR* (**bcTol-M**), is synthesized from **31**. Along with this biradical, asymmetric **Cyolyl-TOTAPOL** biradical **38** containing five hydroxy groups is obtained (Scheme 7) [107]. The synthesis of the latter is carried out by the nucleophilic opening of epoxide **37**, prepared from 4-hydroxy derivative **36** via a reaction with epichlorohydrin and in interaction with amino derivative **34**. Moreover, heterobiradical **AsymPolPOK** (**43**), one of the best DNP agents applied at magnetic fields both 9.4 and 18.8 T, has also been derived from SNR **34**. Thus, the coupling of amine **34** with paramagnetic carboxylic acid **39** produces a heterobiradical, amide **40**, in a moderate yield. Deprotection of the latter and the phosphorylation of intermediate diol **41** with bis(2-cyanoethyl)-*N*,*N*-diisopropylphosphoramidite (BCEDIPPA) in the presence of an activator (5-(benzylthio)-1*H*-tetrazole; BTT) affords diphosphonate **42**. The removal of cyanoethyl groups from **42** under base-catalyzed conditions quantitatively generates the water-soluble dipotassium salt of bis(phosphonate) **43** (Scheme 7) [60]. 

In addition to cyclohexanone derivatives, other cyclic ketones have been reacted with acetonine **5**, for example, tetrahydro-4*H*-pyran-4-one **44** and its thio analog, tetrahydro-4*H*-thiopyran-4-one **45** (Scheme 8) [108]. The yields of cyclic amines in both reactions are approximately the same. Nonetheless, for the pyran derivative, the oxidation proceeds in accordance with a standard scheme and gives SNR **46** with a high yield, whereas in the case of thiopyran compound **47**, this oxidation is accompanied by additional oxidation of the sulfide group to sulfone, resulting in a rather low yield of final SNR **48** [109]. Of note the thiopyran ring can serve as a protective group for the ethyl moiety; consequently, amine **47** has gained popularity in the synthesis of 2,2,6,6-tetraethyl derivatives of 4-oxypiperidine-1-oxyl (TEMPON) **49**, which are NRs extremely resistant to the action of various reducing agents (Scheme 8).

An alternative route for the synthesis of mono-α-SNR **50** and di-α,α′-SNR **20** was suggested [110]. As a result of an aldol condensation of mesityl oxide with cyclohexanone, ketol **51** is converted into a mixture of isomeric ketones **52a,b**. Its treatment with gaseous ammonia in an autoclave at temperatures above 100 °C for 17 h leads to cyclic amine **21b** with a good yield (Scheme 9). Similarly, a mixture of ketones **54a,b** that is obtained by condensation of acetone with cyclohexanone followed by dehydration of **53** is again applied to the same reaction with cyclohexanone, including distillation and dehydration of **55a,b**, thus affording a mixture of three ketones **56a–c**. Finally, the treatment of the latter with ammonia and the oxidation of dispirocyclic amine **22b** produces the desired SNR **20**. It must be noted that although the H_2_O_2_/Na_2_WO_4_ system can be utilized for successful oxidation of amine **21b**, in the case of more sterically hindered amine **22b**, a satisfactory result is achieved only with *m*-CPBA (Scheme 9).

Yamada et al. proposed an alternative approach to the synthesis of piperidine-type SNRs. It is assumed that the carbonyl group in 1,2,2,6,6-pentamethylpiperidine-4-one **57** can undergo enolization, thus simplifying further recyclization of the enol triggered by a carbonyl compound. For instance, heating a solution of compound **57** in DMSO with cyclic ketones **58a–f** in the presence of a large excess of NH_4_Cl provides spirocyclic amines **22b**, **47**, and **59b,d–f** with modest yields. Oxidation of the above amines by hydrogen peroxide in ethanol leads to respective SNRs **20**, **46**, **48**, and **60d,f** [52,109]. In addition, the cleavage of the dioxolane protective group in **59f**, followed by oxidation of intermediate amine **61**, has allowed the preparation of α,α′-di-SNR **62** with three keto groups (Scheme 10) [109].

Spin-labeled nitroxides **63a–c** have been obtained by the aforementioned synthetic procedure, using ^15^NH_4_Cl as a catalyst at the step of amine formation (Scheme 10). This result proves that ammonium chloride is the source of nitrogen at the recyclization and amine formation steps. Because amine **57** is a stronger base than NH_4_Cl, their interaction causes ammonia formation. A plausible scheme of the reaction between **57** and ketones has been proposed. Initiating the whole process is aldol condensation of **57** with cyclic ketone **58** yielding sterically strained **64**, followed by C–C bond cleavage through Grob-type fragmentation [111] providing α,β-unsaturated carbonyl compound **65**. Interaction of the latter with ammonia and a subsequent reaction of **66** with cyclohexanone generates intermediately **67**; further elimination in it, accompanied by cycle closure in aminoenone **68** finally forms amine **59** (Scheme 11) [109]. Yamada et al. noted that, with a decrease in the molar ratios of NH_4_Cl and cyclic ketone **58** to piperidone **57** and at a decreased temperature of the reaction mixture, amines of type **7** bearing only one spiro moiety can form in some cases (ca. 10–30% yield), and their oxidation leads to corresponding SNRs [52,112].

Paramagnetic unnatural amino acids—7-aza-dispiro[5.1.5.3]hexadecane-7-oxyl-15-amino- 15-carboxylic acid **69** and its *N*-(9-fluorenylmethoxycarbonyl) derivative **70**—have been obtained from the above-mentioned SNR **20** to be applied as spin labels. Thus, a reaction of radical **20** with (NH_4_)_2_CO_3_ in the presence of sodium cyanide produces hydantoin derivative **71** with a high yield. Hydrolysis of **71** under harsh conditions—similar to what has been conducted before for the corresponding tetramethyl analog of piperidine-1-oxyl-4-amino-4-carboxylic acid (TOAC) (**72**) [113]—leads to amino acid **69** in a ~50% yield. Nonetheless, the sample obtained in this way has failed to be fully rid of paramagnetic traces and the hydrolysis intermediate. It is assumed that the steric hindrances produced by cyclohexane rings make the complete conversion of the hydantoin intermediate to α-amino acid **69** difficult. In this regard, SNR **69** was synthesized in two steps; first, double Boc-protected derivative **73** is obtained followed by its hydrolysis under relatively mild conditions. Amino acid **69** is then modified at the amino group via interaction with *N*-(9-fluorenylmethoxycarbonyloxy)succinimide, affording target compound **70** without any paramagnetic impurities (Scheme 12) [24].

It is noteworthy that on the basis of a simple paramagnetic amino acid, TOAC (**72**, a derivative of TEMPO), using alternating protection of functional groups in the radical, a dipeptide is obtained that thermolytically and readily converts into spirocyclic nitroxyl biradical (SNBR) **74** of the TEMPO type, containing an almost flat diketopiperazine linker between the paramagnetic nuclei (Figure 5).

Thus, by a previously developed method [114], one part of TOAC is protected at the amino group by a fluorenylmethyloxycarbonyl residue, whereas the carboxy function of another part is esterified with diazomethane. Then, both molecules **75** and **76** are crosslinked under the conditions of peptide bond formation, and after the removal of the fluorenylmethoxycarbonyl (Fmoc) group in **77** by means of diethylamine, aminoester **78** is subjected to mild acid-catalyzed thermolysis accompanied by cyclization and formation of final SNBR **74** (Scheme 13) [115].

Modern application of the DNP method allows us to increase NMR signal intensity in solids and liquids by several orders of magnitude. The DNP method is based on polarization transfer from paramagnetic compounds to nuclei of investigated diamagnetic samples by microwave irradiation of the sample at the electron paramagnetic resonance (EPR) frequency. In practice, a huge variety of paramagnetic compounds has been synthesized and applied as DNP agents during the last decade. The best results on signal enhancement have now been shown by sterically hindered spirocyclic nitroxyl biradicals (SNBRs) of the TEMPO series, which are fairly well soluble in protic solvents and carry various substituents in the spiro substituent, which allows the surroundings of the paramagnetic center to adopt a favorable conformation (see Figure 2) [63]. To obtain promising DNP agents, two approaches were recently used to synthesize SNR biradicals. These methods differ in only one fundamental characteristic, i.e., the nature of the linker between the nitroxide nuclei, in particular, and whether it is flexible or rigid [59]. Some examples from both approaches are presented below.

A urea residue is often employed as a flexible hydrophilic linker. For instance, water-soluble polarizing agents **PyPol** and **AMUPol** are synthesized, starting from SNR **46**. Reductive amination of **46** using ammonium acetate (or tetra(ethyleneglycol) methyl ether amine) and sodium cyanoborohydride or Na(OAc)_3_BH leads to amines **79a,b**, which are then reacted with triphosgene to obtain **80a** (**PyPol**) or **80b** (**AMUPol**) as solids (Scheme 14) [116].

Another popular spacer between two radical parts, methyleneamine, which has one group less than the urea residue, was recently used as the basis for the **TinyPol** family of biradicals [62]. First, from SNR **46**, 4-cyano derivative **81** was prepared from tosylmethylisocyanide in the presence of potassium *tert*-butoxide. Hydrolysis of the cyano group and subsequent reduction of the carboxylic group resulted in primary alcohol **82** in a good yield. Oxidation of the hydroxy group led to formyl-substituted SNR **83**, which then reacted with another SNR, amine **34**, via reductive amination. Finally, deprotection of the silyl groups in an intermediate adduct by means of tetra-*n*-butylammonium fluoride (TBAF) in tetrahydrofurane (THF) yielded **TinyPol** (**84**) as a red solid (Scheme 15).

The synthesis of similar SNR biradicals with a flexible linker was also presented above, in Scheme 7 (compounds **35** and **38**).

In strong magnetic fields, the effective electron resonance frequency, which provides the polarization of unpaired electrons of the biradical, can depend substantially on the molecular orientation relative to the external magnetic field, and in a biradical, the respective parameter is controlled by relative orientations of electron g-tensors. A rigid linker that fixes the two TEMPO moieties in a desired relative orientation should increase the enhancement afforded by the polarizing agent. The required orthogonality of paramagnetic centers toward each other—and their mutual arrangement that prevents a strong exchange interaction—can be realized, for example, in the case of a rigid γ,γ′-SNBR of the TEMPO type in which the number of spirocycles linking the paramagnetic cores is an even number [42]. In this regard, when searching for optimal DNP agents, some investigators have published a number of studies on the synthesis of similar SNBRs.

The reaction of 2,2,6,6-tetramethyl-4-piperidone **6** or substituted piperidin-4-ones **7a,b**, **22b**, **47**, **59b,e**, or **85a–e** with pentaerythritol in the presence of PTSA yields diamines **86a–l**, which are then oxidized with hydrogen peroxide in the presence of Na_2_WO_4_ to trispirocyclic biradicals **87a [42]**, **87b** [45], **87c–f** [52], or **87g–l** [59] in 35–85% yields (Scheme 16).

To carry out cross-condensation and obtain an asymmetric biradical, a mixture of spirocyclic amine **22b**, piperidone **6**, and pentaerythritol was refluxed in toluene with PTSA, followed by the oxidation of the resulting mixture of three amines. Along with the cross-condensation product, SNBR **88**, whose yield is only 10%, symmetric dimeric SNBR **87a,b** are isolated by chromatography [52].

A series of sulfur-containing SNBRs was synthesized in the search for water-soluble biradicals suitable for DNP experiments in water [46,47]. For example, boiling of tetraacetyl pentaerythritithiol **89** with 2,2,6,6-tetramethyl-4-piperidone **6** in concentrated hydrochloric acid yields a hydrochloride of the condensation product in a high yield, whose oxidation with ruthenium tetroxide (generated in situ from RuCl_3_ and H_5_IO_6_) leads to diamine tetrasulfone **90**. Oxidation of the latter with *m*-CPBA gives rise to bisnitroxide **91** in a 70% yield. This SNBR **91** does not have sufficient solubility in water but is readily soluble in the DMSO/H_2_O system with a ratio of 60:40 (Scheme 17) [46]. Piperidin-4-one **59b** also condenses with tetraacetyl pentaerythritiol **89** in HCl, with diamine **92** forming in a 70% yield. Its subsequent exhaustive oxidation to tetrasulfone **93** is performed when potassium peroxomonosulfate is used. Initially, at low pH levels, only sulfur atoms are oxidized; with an increase in pH to 8–9 and oxidation of the mixture with dimethyl dioxirane (generated from acetone added to the mixture), the secondary amino groups are converted into nitroxyl ones. To the disappointment of the researchers, biradical **93** has proven to be also insoluble in glycerin–water mixture. Finally, oxidation of amine **92** with an excess of *m*-CPBA yields a mixture of sulfur-containing bisnitroxides **94**, which have turned out to be quite soluble in a glycerol–water (60:40) mixture (Scheme 17) [47].

## 3. Benzoannelated Derivatives of Piperidine-Type SNRs

### 3.1. SNRs of 1,2,3,4-Tetrahydroquinolines Series

Several examples of the synthesis of stable SNRs of the 1,2,3,4-tetrahydroquinoline series are described in the literature. Their preparation includes two steps: (a) key assembly of the tetrahydroquinoline frame as a result of an “unusual” Diels–Alder cycloaddition of Schiff base **96** to 2-methyl-4,5-dihydrofuran in the presence of boron trifluoride etherate as a catalyst [117] and (b) oxidation of the obtained cyclic amine with hydrogen peroxide to a nitroxyl radical. For example, the interaction of imines **96a,b** with dihydrosylvan produces corresponding tetrahydroquinolines **97a,b**. Their oxidation in the 30% aqueous H_2_O_2_/Na_2_WO_4_ system leads to SNRs **98a,b** [118,119]. Cyclohexylidene derivatives have been obtained in a similar way from α- and β-naphthylamines **99**, which, as a result of similar transformations, are converted into SNRs **100** and **101**, i.e., stable radicals of the benzoquinoline series (Scheme 18) [120].

In a later work [121], precise determination of the structure for the product of the reaction between unsubstituted ketimine **96a** and dihydrosylvan was performed by X-ray analysis, and it was shown that in this process, compound **102** is formed, not compound **97a**, as was postulated earlier [117]. Adduct **102** appears to be formed by the reaction of imine **96a** with the isomeric form of dihydrosylvan, 2-methylenetetrahydrofuran, although usually, the equilibrium between these two tautomers is shifted toward 2-methyl-4,5-dihydrofuran. Thus, in the case of unsubstituted imine **96a**, the reaction proceeds in an unusual way; according to the authors of that study, this outcome can be explained by the influence of steric factors during the formation of the transition state. According to X-ray diffraction data, stable SNR **103** that is obtained by oxidation of amine **102** has the structure of dispirocycle **103** with spiro substituents at α- and γ-carbon atoms, not of mono-spiro compound **98a**, as previously stated. Of note, SNR **103** has been used as an effective indicator of peroxy radicals, with which it reacts and yields a characteristic product (absorption band at 371 nm) in autoxidation systems [122].

### 3.2. SNRs of the 10H,10′H-9,9′-Spirobi[acridine] Series

Diradicals with the orthogonal arrangement of single occupied molecular orbitals (SOMO) have received much attention in the chemistry of organic magnetic materials because they have a ground triplet state, and consequently, the implementation of an intramolecular ferromagnetic interaction in them is possible. At the beginning of the 21st century, several reports appeared about obtaining and studying the magnetic characteristics of spirocyclic diradicals of the acridine type, whose structural features hold promise for achieving orthogonality of closely spaced paramagnetic centers. Indeed, the diamagnetic precursor of such a diradical, 9,9′(10*H*,10*H*′)-spirobiacridine **104**, is synthesized in two stages—from *t*-butyloxycarbonyl (Boc)-protected diphenylamine and methoxyethoxymethyl (MEM)-protected acridone. The presence of Boc-protection in the starting amine promotes its directed *ortho*-lithiation to intermediate **105**. The key stage includes a one-pot procedure consisting of the addition of a heteroaromatic ketone to *ortho*-lithiated compound **105**, followed by the acid-catalyzed intramolecular Friedel–Crafts reaction and final acid-promoted removal of the protective groups to obtain target spirocyclic diamine **104** with a good yield relative to starting diphenylamine [123]. The oxidation of diamine **104** by *m*-CPBA leads to the formation of a mixture of monoradical **106** (35%) and target diradical **107** (5%), which are separated by chromatography (Scheme 19). The structure of both radicals has been proved by X-ray analysis (Figure 6). A study on the magnetic properties of diradical **107** in a polycrystalline sample has shown that antiferromagnetic interactions are predominant for this compound. This result was explained by the authors as a possible mutual influence of the closely located nitroxyl groups of neighboring molecules. Nevertheless, the EPR spectrum of a dilute solution of compound **107** and Density Functional Theory (DFT) calculations in the gas phase indicated that the ground state of the diradical molecule should be a triplet [78].

A decade later, the investigation of Ishida et al. was successful; recently, a publication appeared on the synthesis of paramagnetic spirocyclic nitroxyl diradical (SNDR) **108** with the largest value of ferromagnetic exchange in the series of spiro compounds, 2*J*/*k*_B_ = + 23(**1**) K [79]. Crystallographic analysis clarified the *D*_2*d*_ molecular structure, suggesting the degeneracy of SOMOs. The introduction of four *tert*-butyl groups at the *para*-position of the benzene cycle in the biacridine system relative to the NO fragment (Scheme 20) has made it possible to completely suppress the effect of the intermolecular antiferromagnetic interaction of the diradical molecules. The key base compound in a six-step sequence, di-*tert*-butyl–substituted acridone **109**, is obtained quantitatively by Friedel–Crafts alkylation of acridone [124]. In a 2020 paper, the same authors showed that, by changing the number of *tert*-butyl groups (0 → 2 → 4) in the molecule of a spirobiacridine diradical, it is possible to control the exchange interaction of unpaired electrons in the molecule by fine-tuning the intermolecular distances [125]. The corresponding di-*tert*-butyl diradical is obtained by a similar procedure from diarylamine **110** and MEM-protected acridone as starting blocks.

## 4. Piperazine- and Morpholine-Type SNRs

Sterically hindered SNRs of the piperazine series (**115, 116, 118**, and **120)** were obtained for the first time to study the effect of substituents on hyperfine coupling constant A_N_ in the EPR spectra of stable radicals [126]. For instance, under reduced pressure, heating the aminonitrile of cyclohexanone **111** gives bis(1-cyanocyclohexyl)amine **112**, followed by a one-pot step, including acid-catalyzed hydrolysis and intramolecular cyclization, leading to the starting compound of SNR synthesis—cyclic piperazinedione **113** [127]. Carboxylation of the latter with chloroethyl formate and oxidation of amine **114** with *m*-CPBA give nitroxide **115** in a 63% yield. Hydrolysis at room temperature and decarboxylation of SNR **115** allows for the isolation of NH-containing radical **116** (Scheme 21). In contrast, using protection, carbonyl groups are removed from the heterocycle. Benzylation of amine **113**, followed by oxidation of intermediate **117** with *m*-CPBA, results in nitroxide **118** in a 61% yield. Reductive deoxygenation of **118** with LAH leads to diamagnetic hydroxylamine derivative **119**, which readily oxidizes in ambient air to radical **120** (Scheme 21).

In the works of Lai, a different approach to the synthesis of SNRs of the piperazinone and morpholinone series was proposed [128,129,130,131]. The key stage of this approach is the Bargellini reaction [132], which consists of the interaction of 1,2-aminoalcohols or 1,2-diamines with ketones in chloroform in the presence of a strong base, with the formation of sterically hindered morpholinone or piperazinone **121**, which can then be oxidized to corresponding nitroxyl radical **122** by the action of 1–2 equiv of *m*-CPBA (Scheme 22).

The following scheme represents the proposed mechanism of the Bargellini reaction. In the presence of a strong base, chloroform is deprotonated to the CCl_3_^−^ anion, which attacks the carbonyl carbon of the ketone, forming dichloroepoxide **123**. A sterically hindered C–N bond is formed through the regioselective opening of **123** with a nucleophile—amino alcohol or diamine. Subsequent cyclization gives rise to lactone or lactam **124**, respectively (Scheme 23).

Below, we provide examples of specific syntheses using the above-mentioned approach to obtain SNRs of the piperazine and morpholine series. For example, condensation of 2-nitropropane, formaldehyde, and optically active (*S*)-1-phenylethylamine produces nitroamine **125**, subsequent reduction of which by hydrogen on Raney nickel quantitatively leads to diamine **126**. The Bargellini reaction of **126** with cyclohexanone causes the formation of intermediate piperazinone **127**. Although the yields in this reaction are high (>80%), in the early work of Lai [128], it was noted that, along with piperazinone **127**, the formation of minor regioisomer **128** is also sometimes observed, while the ratio of products **127/128** is ~4.5:1. Oxidation of cyclic amine **127** with *m*-CPBA generates optically active radical **129** in a quantitative yield (Scheme 24) [133].

Similarly, hydrogenation of the intermediate obtained by the condensation of 1-nitrotetralin with formaldehyde and (*S*)-phenylethylamine gives diamine **130**. Its reaction with acetone under the conditions of the Lai method and oxidation of piperazinone **131** quantitatively produces optically active nitroxide **132**. Reduction of lactone **131** with LAH yields piperazine **133** in the form of a mixture of diastereomers in a 1.3:1 ratio, which are efficiently separated by crystallization of salts with camphorsulfonic acid (CSA). Hydrogenation of piperazine **133** on Pd(OH)_2_ results in the removal of the phenylethyl group, and subsequent treatment with 1 equivalent of TsCl yields mono-tosyl derivative **134**. Optically active nitroxide **135** is obtained quantitatively by the oxidation of the NH-function in **134** (Scheme 25) [133].

Chiral morpholine and morpholinone nitroxides should also be available using the Bargellini reaction. Scheme 26 outlines a route to these nitroxides from model 2-methyl-2-amino-1-propanol. Investigators were surprised to find that the Bargellini coupling with acetophenone works very well in this case. The same reaction has been unsuccessful with diamine **130**. Sodium salt **136** precipitates from the reaction mixture and is isolated by filtration. Acidification, followed by heating in toluene and neutralization with Et_3_N, produces lactone **137** in a 57% overall yield from the initial amino alcohol. Oxidation with *m*-CPBA completes this simple and effective synthesis of stable morpholinone nitroxide **138**.

The next step was the synthesis of an optically active SNR of the morpholine series, for which amino alcohol **139** (obtained by the condensation of 1-nitroindane with formaldehyde, followed by reduction of the nitro group) was used as a starting compound. The three-component reaction of compound **139** with acetone in chloroform in the presence of NaOH yields the corresponding carboxylate, whose acidification and heating in PhMe result in lactone **140** in a 66% yield. Separation of the enantiomers of morpholinone **140** is again carried out by crystallization of their salts with (+)-camphorsulfonic acid. The diastereomeric mixture of acetates **141** is obtained by the reduction of the (+)-**140** isomer with diisobutylaluminium hydride (DIBAL-H), followed by acylation. *O*-acyl derivative **141** is transformed by means of trimethylsilyl trifluoromethanesulfonate (TMSOTf) and trimethyl(phenylthio)silane (TMSSPh) into *S*-derivative **142**, which is further reduced with Li/NH_3_ to morpholine. Controlled oxidation of the latter with *m*-CPBA acid in the presence of NaHCO_3_ in a two-phase DCM/H_2_O system leads to optically active radical (+)-**143** with a very high yield (Scheme 27) [133]. Of note, overoxidation (overexposure to an oxidizing agent) of radical **143** causes a complete loss of optical activity because of rapid racemization arising from the recyclization of the spiro function as a consequence of solvolytic opening and closure of the *N*-oxoammonium salt, which is a product of nitroxide overoxidation.

On the basis of a natural ketone, *L*-(−)-menthone, Studer et al. performed the synthesis of highly sterically hindered SNRs for subsequent preparation of the corresponding alkoxyamines to be applied as initiators/regulators of controlled nitroxide-mediated radical polymerization of *n*-butyl acrylate and styrene. The key compound in the scheme for synthesizing these SNRs is hydantoin **144**, which is obtained via the Bucherer–Bergs reaction [134]. Its further hydrolysis under harsh conditions, followed by treatment of the intermediate amino acid with isopropylamine in the presence of a condensing agent and hydrogenation with BMS, or the use of reductive conditions (NaBH_4_ / I_2_), leads to diamine **145** and amino alcohol **146**, respectively. The Bargellini reaction of **145** with cyclohexanone yields dispirocyclic piperazine **147**, whose final stage of oxidation to target SNR **148** is carried out in 40% peracetic acid (Scheme 28) [73].

On the contrary, for the synthesis of the chiral SNR of the morpholine series, **153**, a new approach was employed, including the use of a commercially available synthetic block, 3-oxetanone [135]. Thus, amino alcohol **146** is reacted with a strained ketone to form dispirocompound **149** in a high yield. In the presence of trimethylsilyl cyanide (TMSCN) and a catalytic amount of In(OTf)_3_, **149** is then subjected to the Strecker reaction with subsequent ring expansion to obtain morpholine **150**, which is isolated as a single diastereoisomer. Nitrile reduction, silyl ether cleavage, and BOC protection result in bicycle **151**. Subsequent alkylation of **151** with an excess of MeI causes double methylation. Again, the oxidation of secondary amine **152** with AcOOH affords the target bulky spiro morpholine-based stable radical **153** (Scheme 28) [74].

These chiral nitroxides carry spiroanellated six-membered rings at the α-position with two additional substituents, which lock a sterically more hindered chair conformation. X-ray structural analysis and DFT calculations have confirmed the conformation of such nitroxides.

A very recent paper presents the successful synthesis of rigid SNR diradicals **154a-c** of the morpholine type for DNP that have small distances between paramagnetic centers and torsion angles that are not accessible in the six-membered SNR biradicals, **bTbK**, and **TEKPOL** series [136]. Their synthesis is based on the original methodology of tin amine protocol (SnAP)-assisted iterative assembly of polyspirocyclic *N*-heterocycles developed by Bode et al. [137]. The key step in the process is the coupling of the cyclic ketone **155** with the SnAP reagent **156** in the presence of a Lewis acid, titanium isopropoxide, and a catalyst (copper (II) triflate), yielding an interaction of the forming carbon-centered radical with a carbon atom of the iminium cation, thus forming a morpholine ring (Scheme 29, Figure 7) [138].

## 5. 2,5-Dihydropyrrole (3-Pyrroline)- and Pyrrolidine (PROXYL)-Type SNRs

One of the main methods for obtaining 3-pyrroline and pyrrolidine NRs is an approach based on the Favorskii rearrangement of 3,5-dibromo-4-oxo-2,2,6,6-tetramethylpiperidine **158** by means of various *O-* and *N*-centered nucleophiles: OH^−^, MeO^−^, NH_3_, or amines. The resulting NH-pyrrolines **159** can be then oxidized to pyrroline nitroxides **160** or reduced to pyrrolidines **161** and then transformed into corresponding pyrrolidine (PROXYLs) **162** by oxidizing agents based on hydrogen peroxide [139]. As for the Favorskii rearrangement of the monobromo derivative 3-bromo-4-oxo-2,2,6,6-tetramethyl piperidine-1-oxyl **163**, by using nucleophiles OH^−^, EtO^−^, and RNH_2_, it directly leads to PROXYLs **162** (Scheme 30) [139].

Thus, piperidine-type SNRs can be easily transformed into SNRs of the pyrrole series; this is a very convenient method for synthesizing the latter. Let us consider an example of the application of this approach to the synthesis of various SNRs of the 3-pyrroline series, **166–175**, which are promising spin labels. Stirring of amine **22b** with 4 equivalent of bromine for 24 h readily results in hydrobromide **164** (Scheme 31). Rearrangement of dibromo derivative **164** in a dioxane−aqueous ammonia mixture produces amide **165**, the oxidation of which with H_2_O_2_ in the presence of sodium tungstate generates paramagnetic amide **166**. Due to the low solubility of nitroxide **166** in water, the hydrolysis of the amide group is carried out in a water–alcohol solution of KOH with microwave activation (165 °C), as a result of which carboxylic acid **167**, a starting compound for obtaining spin labels, is synthesized with a good yield [21]. Further, starting with acid **167**, the methanethiosulfonate label is synthesized in five steps, and although nitroxide **172** is poorly soluble in water, it is reported to be capable of reacting with glutathione in aqueous methanol with the formation of SNR **173**, which precipitates from a concentrated aqueous solution as an individual compound (Scheme 31).

Moreover, on the basis of acid **167**, its *N*-hydroxysuccinimide ester **174** is obtained [21], which is applied to the synthesis of paramagnetic unnatural amino acid **175** [140] (Scheme 31) and to a novel site-directed spin labeling (SDSL) approach potentially suitable for long natural RNAs. In particular, site-specific alkylation of an RNA with the 4-[*N*-(2-chloroethyl-*N*-methyl)amino]benzyl phosphoramide derivative of oligodeoxyribonucleotide, followed by the hydrolysis of the phosphoramide bond and a release of the aliphatic amino group in a linker attached to the target nucleotide residue, creates a convenient template for spin-labeling by selective coupling of paramagnetic ester **174** to this amino group [37]. *O*-Mesyl derivative **170** was recently utilized to synthesize an aminomethyl spin label for binding through a flexible linker to C_60_ in order to apply photoexcited fullerenes as spin labels for pulsed dipolar (PD) EPR distance measurements [141].

Pyrrolidine radicals (PROXYLs) are currently some of the stablest and most robust NRs, and as a result, the most suitable for and demanded by scientists; in this regard, many synthetic approaches have been developed specifically for this class of paramagnets. Through the Favorskii rearrangement of monobromo derivative **176**, carboxylic acid **177** is obtained from SNR **20** in two stages—bromination of **20** to halogenated derivative **176** and subsequent ring contraction to form SNR **177**, with yields of ~60% at each subsequent step [82]. At the same time, acid **177** can be obtained by the reduction of pyrroline derivatives. Thus, the interaction of amide **166** with lithium borohydride proceeds selectively only at the double C=C bond without affecting the amide group and radical center. Alkaline hydrolysis of the reduced intermediate in aqueous alcohol leads to SNR **177** in a 67% yield (Scheme 32) [21].

The synthesized acid **177** has been actively applied to the preparation of different spin labels. For example, its succinimide derivative **178** was subjected to the synthesis of water-soluble contrast agent ORCA, a polyradical dendrimer (Figure 8), via a sequential interaction of the amino groups of the polypropyleneimine molecule with the NHS-ester of SNR **178** and then with the succinimide derivative of polyethylene glycol (Scheme 32) [82].

Derivative **178** has also been utilized for the synthesis of isocyanate spin label **179**, which in turn is conjugated with a natural molecular UV filter, kynurenine (KN), to significantly increase its photostability in modified KN **180** (Scheme 32) [142]. Bifunctional derivatives of SNRs–PROXYLs can be prepared by nucleophilic addition to a double bond activated by the presence of an acceptor group in 3-pyrroline. In this way, amide **166** is converted into the corresponding nitrile, followed by the addition of the CN^−^ anion resulting in a mixture of *cis*- and *trans*-dicyano derivatives **181**. Both dinitriles were separated and subjected to alkaline hydrolysis in a drastic conditions (10% aq KOH, 100→150 °C, 6 d, conversion 63%), resulted in a formation of solely one product, *trans*-dicarboxylic acid **182** [21].

A rather unusual method for synthesizing SNR of the PROXYL series was demonstrated by Motherwell and Roberts. Spiro-nitroxide **183** is prepared by oxidative cyclization of enehydroxylamine with a silver compound. Thus, the Michael addition of methyl vinyl ketone to nitrocyclohexane **184** followed by the Wittig reaction with CH_2_=PPh_3_ leads to γ-nitroolefine **185**. Reduction of the nitrocompound by Zn/NH_4_Cl and subsequent oxidation of enehydroxylamine **186** by Ag_2_CO_3_/celite or Ag_2_O allows us to obtain the target nitroxide **183** in a 30% yield. The authors suggested that the precursor of SNR **183** is open-chain monoalkyl nitroxide **187**, which can undergo cyclization similarly to a certain olefinic aminium radical cation (Scheme 33) [143].

Two more general approaches to the synthesis of NRs of the PROXYL series, based on pyrroline nitrones, have been developed by Keana and Hideg groups.

In the first method, the key step is the Grignard addition to nitrone **188** (R_1_ = H). The resulting *N*-hydroxy intermediate is oxidized to nitrone **189**, which is reacted with another Grignard reagent; this time, the oxidation of the intermediate generates stable PROXYL **190** (Scheme 34) [144]. In another method, the crucial stage is a regioselective reaction of 1,3-dipolar cycloaddition of nitrone **188** (R_1_ = Me) to an alkene activated by an acceptor group thereby forming isoxazolidine **191**. Bicycle cleavage at the N–O bond by the Zn/AcOH reducing system and subsequent oxidation of amine **192** yields the corresponding NR **193** (Scheme 34) [145]. Next, the application of these two approaches to the synthesis of the corresponding SNRs of the PROXYL series is demonstrated in a number of examples.

PROXYL type SNRs **198**, **200**, **201**, and **204**, possessing an adamantane part as a spiro substituent, have been obtained with the aim of their subsequent modification and use as spin labels. The base-catalyzed Michael addition of methyl vinyl ketone to 2-nitroadamantane **194** leads to γ-nitroketone **195**, followed by reductive cyclization with Zn/NH_4_Cl to starting spirocyclic pyrroline **196** (Scheme 35).

Nucleophilic addition of methyl or ethynylmagnesium bromides to nitrone **196** leads readily to hydroxylamines **197** and **199**, which are then transformed into corresponding SNRs **198** and **200** by the action of MnO_2_. The reaction of **196** with the Grignard reagent BrMgC≡CCH_2_OMgBr, which is synthesized from propargyl alcohol and two equivalents of EtMgBr, leads to PROXYL **201** through the formation of an intermediate, hydroxylamine **202** (Scheme 36).

1,3-Dipolar cycloaddition of methyl acrylate with nitrone **196** occurs in an excess of the reagent upon heating in the absence of a solvent and results in isoxazolidine **203** in a 76% yield. N–O bond breakage in bicyclic compound **203**, by heating in the Zn/AcOH system, allows us to obtain a sterically hindered amino alcohol, which, upon treatment with *m*-CPBA, is converted to corresponding SNR **204**, albeit with a low yield (Scheme 36) [22].

It is worth noting that the use of a tandem Michael reaction of methyl vinyl ketone with cyclic nitro derivatives, followed by the addition of an organometallic reagent to a five-membered nitrone, is a fairly general method for constructing SNRs of the PROXYL series. In this way, Tamura et al. used the approach described above for the synthesis of mesogenic paramagnetic compounds containing a spirocyclic residue. For example, nitrocyclohexane **184** quantitatively forms adduct **205** in a reaction with methyl vinyl ketone in the presence of tetramethylguanidine (TMG). Its subsequent reduction under classic conditions (Zn/NH_4_Cl/EtOH) generates the product of intramolecular cyclization of an intermediate hydroxylaminoketone, spirocyclic nitrone **206**. The addition of arylmagnesium bromide to heterocycle **206**, followed by oxidation and desilylation without isolation of intermediates, affords target SNR **207** in a 42% yield per nitrone **206** (Scheme 37) [146].

In continuation of this work, to obtain new all-organic ferromagnetic liquid crystal materials, a synthetic scheme based on 1,4-dinitrocyclohexane has been successfully implemented, which gives rise to *cis*- and *trans*-isomers of PROXYL-type α,α′-SNDRs **213**, in which paramagnetic nuclei are separated by a rigid spirocyclic cyclohexane linker [147]. Thus, 1,4-cyclohexanedioxime **208** is oxidized in one step with a satisfactory yield to 1,4-dinitro derivative **209** using the novel system Na_2_MoO_4_/H_2_O_2_/MeCN/H_2_O. A reaction of **209** with methyl vinyl ketone, followed by reduction of adduct **210**, produces a mixture of *cis*- and *trans*-isomers of spirocyclic dinitrone **211** in a ~3:2 ratio, which are successfully separated by chromatography. The addition of a Grignard reagent based on protected *para*-bromophenol to dispirocycle **211** at −78 °C proceeds stereoselectively, albeit in a low yield, and the resulting bishydroxylamines are oxidized without isolation to corresponding diradicals *cis*-**212** and *trans*-**212** (X = OTBDMS). Final desilylation almost quantitatively leads to bisphenols, SNDRs *cis*-**213** and *trans*-**213** (Scheme 38). For the synthesis of a paramagnetic discotic compound, *trans*-diradical **214**, dinitrone **211** is treated with an excess of 4-(diethoxymethyl)phenylmagnesium bromide in THF at ambient temperature for 24 h; in this case, the yield of corresponding SNDR **212** (X = CH(OEt)_2_) increases up to 37%. Subsequent hydrolysis of *trans*-acetal **212**, selective oxidation of diformyl derivative **215** to paramagnetic dicarboxylic acid **216**, and the final amidation step involving amine **217** and appropriate condensation reagents (DMT-MM and *N*-methylmorpholine) result in α,α′-SNDR **214** (Scheme 38), showing both discotic and smectic liquid crystalline phases and manifesting superparamagnetic-like behavior [81,148]. 

Another similar approach to the synthesis of SNRs of the PROXYL type, which contain an easily modifiable 1,3-dioxane moiety in the spiro substituent, was implemented in the work of Japanese researchers in 2019. Based on 2-bromo-2-nitropropane-1,3-diol **218**, 2,2-dimethyl-5-nitro-1,3-dioxane was synthesized, and then consecutively by Michael and Grignard additions, it was transformed through intermediate nitrone **219** into 5-R-substituted SNRs **220a-g** (Scheme 39) [149]. The study also described the physical properties of synthesized SNRs **220**, e.g., log P and water−proton relaxivity (*r*_1_, a measure of the ability to serve as a contrast agent), and the reactivity (*k*_2_) of these species toward ascorbic acid (AsA) was evaluated. An acetal ring in SNRs **220** is capable of deprotection under acidic conditions, to form a diol analog with increased *r*_1_; thus, SNRs might enhance contrast imaging and tumor detection using these radicals might be a valid technique.

Chiral super sterically hindered SNR **221** is obtained by multistep synthesis via reactions of the addition of organomagnesium reagents to cyclic nitrones and intramolecular 1,3-dipolar cycloaddition of nitrones containing a terminal alkene group. (3*S*,4*S*)-3,4-Di-*tert*-butylpyrroline *N*-oxide **222** has been chosen as a starting compound because of its unique ability to enter into 1,3-dipolar cycloaddition reactions even with unactivated alkenes. It has been obtained previously in a good yield from the (*R*,*R*)-*O*,*O*′-di-*tert*-butyl ester of tartaric acid **223** as a result of simple four-step synthesis, including hydroxylamine-promoted cyclization of a diol ditosylate compound (Scheme 40) [150].

The addition of 4-pentenylmagnesium bromide to nitrone **222** proceeds stereoselectively due to the steric effect of the bulky *tert*-butoxyl group at the third position on the pyrroline ring and causes the formation of *N*-hydroxypyrrolidine **224**. Oxidation of the latter proceeds with moderate regioselectivity and is accompanied by the emergence of two isomeric nitrones **225** and **226**, which are separated by column chromatography. Intramolecular cycloaddition in nitrone **225** is performed at 110 °C for 2.5 h and results in a single product, tricycle **227**, in a quantitative yield. Treatment of compound **227** with *m*-CPBA opens the isoxazolidine part and produces aldonitrone **228**, which is again subjected to a sequence of addition/oxidation reactions with quantitative formation of nitrone **229**. The second regioisomer, aldonitrone **226**, was also converted into spiro compound **229** during five-step transformations (**226**→**230**→**231**→**232**→**233**→**229**), including Grignard addition, oxidation, intramolecular (3 + 2)-cycloaddition, reductive cleavage of the isoxazolidine ring promoted by an low-valent titanium (LVT)-reagent, and final oxidation of the amino alcohol with H_2_O_2_/Na_2_WO_4_ (Scheme 40).

Transformation of nitrone **229** into a tricyclic product proceeds in a stereospecific manner leading exclusively to the formation of a single cycloadduct, **234**, whose spatial structure has been confirmed by NMR spectroscopy. Again, treatment of **234** with a low-titanium reagent produces aminodiol **235**, followed by final oxidation to a corresponding enantiomerically pure dispirocyclic radical, whose structure has been proved by X-ray analysis (Figure 9). The synthesized SNR **221**, due to its very high kinetic stability (determined mainly by steric factors), shows exceptional inertness toward biological reductants [23].

Later, the same researchers suggested that the presence of electron-withdrawing *tert*-butoxy groups at positions 3 and 4 of the pyrrolidine ring of **221** can strongly affect the rate of nitroxide reduction; therefore, the removal of these groups should cause a further decrease in the reduction rate of the corresponding nitroxide. In this regard, they developed a method for the synthesis of SNRs of *C*_1_-symmetric racemic 3,4-unsubstituted pyrrolidine nitroxides **236a–c** with only one spiro (2-hydroxymethyl) cyclopentane moiety (Scheme 41) [151]. Before the step of oxidation of the secondary amino group to a radical in this sequence, acyl protection of the primary alcohol group (**237**→**238**) is carried out. In one case (**238a,** R = Me), this leads to the formation of a byproduct, SNR **239**, which has a double bond in the spirocycle. In the authors’ opinion, the dehydrogenation reaction with the formation of such an unusual radical can be explained by the participation of oxammonium cation **240**, a probable intermediate of the oxidation reaction.

Earlier, a curious example of the preparation of PROXYL-type nitroxide **241**, spiro-fused with an isoxazoline heterocycle, was described. Namely, a study on nucleophilic reactions of 5,5-dimethyl-2-phenacylpyrroline-1-oxide **242**, which is obtained by the interaction of metalated derivative 2,5,5-trimethylpyrroline 1-oxide **243** with ethyl benzoate, revealed that the treatment of nitrone **242** with hydroxylamine leads to oxime **244**, which can exist in solutions in two tautomeric forms, **244** and **245**, the latter being spirocyclic. Oxidation of compound **245** with MnO_2_ in chloroform produces SNR **241** in a high yield (Scheme 42) [152].

## 6. 2,5-Dihydroimidazole (3-Imidazoline)-Type SNRs

In the 1970s–1990s, Volodarskii et al. developed an approach to the synthesis of nitroxyl radicals of the 2,5-dihydroimidazole series (3-imidazolines); the method consists of the condensation of 2-hydroxylaminoketones (HAKs) **246** with carbonyl compounds in the presence of ammonia or ammonium acetate, followed by conversion of an intermediate, 1-hydroxyimidazolines **247**, to nitroxides **248** by means of heterogeneous or homogeneous oxidants, such as MnO_2_, PbO_2_, or Cu(OAc)_2_/NH_3_/O_2_ [153]. It should be noted that the condensation of HAKs with weakly reactive ketones is accompanied by the formation of dihydropyrazine 1,4-dioxides **249**, i.e., byproducts of the HAK dimerization reaction catalyzed by ammonium acetate (Scheme 43) [154]. When HAKs are heated in the absence of NH_4_OAc, virtually no pyrazine dioxides are formed.

The use of a cyclic ketone as a substrate in this procedure generates an SNR of the 3-imidazoline series. For instance, condensation of HAKs **246** with cyclopentanone, cyclohexanone, or heterocyclic ketones in the presence of excess NH_4_OAc in methanol yields 1-hydroxy derivatives **247a–g,** whose subsequent oxidation with manganese dioxide (or 2,3-dichloro-5,6-dicyano-1,4-benzoquinone (DDQ) in the case of compound **247g**) leads to agents of radical polymerization and paramagnetic ligands, the corresponding SNRs **248a–d [155]**, **e–f [154]**, and **g [156]** (Scheme 43).

HAK **250** (R = H), bearing an additional functional phenolic group, has been employed for the synthesis of SNRs possessing one or two mesogenic groups. For this purpose, compounds **250** (R = H, C_n_H_2n+1_) are condensed with a cycloxexanone [157] or with its four-substituted functional derivatives, 4-(4-hydroxyphenyl)cyclohexanone [158] and 4-hydroxycyclohexanone [159]. After the oxidation of intermediate *N*-hydroxy derivatives to SNRs **251**, the latter are acylated by means of one or two residues of 4-alkoxybenzoic acids to form corresponding paramagnetic esters **252** (one of examples is shown in Scheme 44). Three mixed triradicals **254a–c** with small exchange coupling parameters (*J* ≪ *A*_N_) are obtained on the basis of a coupling reaction between spirofused 2,5-dihydroimidazole–type monoradical **251** and two molar equivalent of carboxylic acid derivatives **253a–c** of PROXYL-, TEMPO-, or 2,5-dihydro-1*H*-pyrrol-type nitroxides [160]. In this case, compound **251** (R = H, X = 4-OH-C_6_H_4_), carrying two phenolic groups with different acidity, serves as a convenient template for the assembly of heteropolyradicals (Scheme 44). The longest spin–spin and spin–lattice relaxation times at 50 K have been documented for a triradical carrying two TEMPO moieties, indicating the potential usefulness of three-spin qubit models for quantum gate operations. 

Symmetric triradicals **256a–c** have been synthesized upon treatment of cyanuric chloride **255** with three equivalent of SNR **251** (X = H; O(C=O)Ar, *p*-C_6_H_4_-O(C=O)Ar) in an attempt to prepare liquid crystalline discotics based on *s*-triazine (Scheme 44). The crystal structure of triradical **256a** has been determined by the X-ray diffraction method [161].

Notably, the reaction of two equivalent of HAK **250** with 1,4-cyclohexandione **157a** does not yield a double-condensation product and the formation of a dispirocycle; instead, it results in a complex mixture of products forms. Therefore, to obtain di-SNDRs of the imidazoline series, a step-by-step strategy is utilized, including either the use of dioxolane protection for the diketone **157a** or the oxidation of the secondary 4-hydroxy group in radical **251** (X = OH) to a keto derivative SNR. It should be noted that the second method has been shown to be more productive, especially when the Dess–Martin reagent is used, whereas the deprotection in the radical **257** has proved to be a challenging task, and only when the system “iron (III) chloride on silicagel” is employed, ketone **258** is isolated with a low yield (Scheme 45). Reduction of the radical function and condensation of the intermediate *N*-hydroxyketone **259** with 2-hydroxylaminoketone **250** (R = H) in an inert atmosphere (followed by oxidation of the reaction product) ensures the isolation of desired diradical **260** in the form of a mixture of *cis*/*trans* isomers in a total yield of 35% across three steps. Acylation of this mixture with 4-alkoxybenzoic acid chloride results in di-SNDRs *cis*-**261** and *trans*-**261**, which are identified as individual isomers after chromatographic separation (Scheme 45). It is reported that the condensation reaction of HAK **250** with diamagnetic hydroxylaminoketone **259** is stereoselective because isomeric nitroxides *trans*-**261** and *cis*-**261** are isolated in a 3.75:1 ratio. EPR spectra of the obtained di-SNDRs stereoisomers **261** confirm their paramagnetic diradical nature and a difference in spatial structure. To determine the spatial structure of the isomers, diradical *trans*-**261** has been reduced by Zn/NH_4_Cl to diamagnetic bis(hydroxylamine), in which the *trans*-arrangement of the heterocycles has been determined by NMR spectroscopy [162]. 

Kirilyuk et al. described a method for the synthesis of SNRs of the 3-imidazoline series based on the reaction of 1,2-ketoximes (isonitrosoketones) with ketones in the presence of NH_4_OAc in acetic acid, followed by the oxidation of the resultant cyclic aldonitrone in methanol with excess PbO_2_ to a stable nitroxide with two methoxy groups on the α-carbon atom near the nitroxyl group. In this manner, from ketoxime **262**, SNR **264a** is synthesized in a high yield by condensation with cyclohexanone and oxidation of intermediate nitrone **263a** (Scheme 46) [163]. Intriguingly, the reduction of radical **264b** obtained according to a similar scheme by means of the Zn/NH_4_Cl system is accompanied by the elimination of the methanol molecule and causes the formation of methoxynitrone **265**. Prolonged exposure of **265** to manganese dioxide in ethylene glycol gives rise to SNR **266**, in which the spiro moiety is a heterocycle of the 1,3-dioxolane type (Scheme 46) [164].

Recently, an approach to the synthesis of a pH-sensitive spin probe based on the SNR of the 3-imidazoline series was proposed, in which the key stages are a sequence of reactions of intramolecular 1,3-dipolar cycloaddition of alkenylnitrones of the 4*H*-imidazole series and an opening of the intermediate cycloadduct by an LVT reagent with the emergence of a spirocyclic hindered amine, whose standard oxidation leads to the target SNR. Thus, at the first stage, a pH-sensitive group is introduced into the molecule together with the components of the dipole and dipolarophile. For this purpose, 1-hydroxy-2,5-dihydroimidazole **267** is synthesized and oxidized (without isolation from the reaction mixture) to 4*H*-imidazole-3-oxide **268**, whose subsequent nitrosation and the Beckmann rearrangement in oxime **269** give carbonitrile **270**. Nucleophilic substitution of the cyano group in the latter leads to 5-dimethylamino-4*H*-imidazole-3-oxide **271** in a high yield. At the second stage, intramolecular 1,3-dipolar cycloaddition of an unactivated C=C bond in ketonitrone is carried out, and its successful implementation is favored by the presence of a suitable linker (cf. [165]). Indeed, heating of **271** in toluene at 110 °C results in a single product, **272**, with a 96% yield. The opening of cycloadduct **272** under the action of an LVT reagent, followed by oxidation of spirocyclic sterically hindered amino alcohol **273** with *m*-CPBA, almost quantitatively yields target SNR **274** (Scheme 47) [29].

The same synthetic approach has been applied by Morozov et al. in another study [72] to obtain sterically hindered mono- and di-SNRs **275a–d** for use of the latter in “living” polymerization. The interaction of aldonitrones **276a–d** with 4-pentenylmagnesium bromide, oxidation of intermediates to ketonitrones, and subsequent thermal intramolecular 1,3-dipolar cycloaddition, produce tricycles **277a**–**d**. The opening of the isoxazolidine ring in **277** is performed using either the Zn/AcOH system or an LVT reagent. The resulting amines are oxidized with *m*-CPBA to target SNRs **275a–d** (Scheme 48).

Paramagnetic 3-imidazoline 3-oxides are obtained by condensation of 2-hydroxylamino oximes **278** with ketones in the presence of NH_4_OAc, where the formed acyclic nitrone **279** exists in a tautomeric equilibrium with the cyclic form—1-hydroxy-3-imidazoline 3-oxide **280**. The latter, when oxidized with manganese dioxide, transforms into a nitroxyl radical, thereby shifting the equilibrium toward the cyclic product. The catalytic effect of ammonium acetate is explained by the formation of a ketone imine at the first step, whose protonated form is more reactive in the nucleophilic attack of the hydroxylamino group. Thus, by this technique, SNR **281** is prepared in a high yield (Scheme 49) [166]. The same SNR **281** is obtained from 2-amino-2-methyl-1-phenylpropan-1-one oxime **282a** under conditions of acid-catalyzed condensation of the latter with cyclohexanone and oxidation of intermediate aminonitrone **283** by hydrogen peroxide (Scheme 49) [167].

Aliphatic aminooxime **282b** can react with triacetonamine only under harsh conditions, whereas the yield of a spirocyclic nitrone, diamine **284**, is rather low. Oxidation of nitrone **284** by H_2_O_2_ leads to a different ratio of diradical **286** to mono-SNR **285**, depending on the duration of incubation. Paramagnetic compounds **285** and **286** are separated by chromatography on silica (Scheme 50) [168].

Another notable type of SNR is exemplified by nitroxyl radicals of the 3-imidazoline 3-oxide series, spiro-fused to the 1,3-dioxolane part at the C-2 atom. Compound **287** is prepared by sequential oxidation of 2,5-dihydroimidazole *N*-oxide **288** with manganese dioxide (using ethylene glycol both as a solvent and reagent) first to hydroxylamine derivative **289** (via the intermediate formation of 4*H*-imidazole *N,N′*-dioxide) and then by oxidation of diamagnetic hydroxylamine **289** by MnO_2_ in chloroform (Scheme 51) [169].

## 7. 4,5-Dihydroimidazole (2-Imidazoline)-Type SNRs

Recently, the vast class of stable nitronyl nitroxyl radicals (NNRs) of the 4,5-dihydroimidazole series, introduced into practice by Ullmann et al. [170,171] and represented mostly as 4,4,5,5-tetramethyl (tetra-alkyl) derivatives, was supplemented with spirocyclic analogs (SNNRs). First, α,β-di-SNNRs were prepared by Ovcharenko et al. via five-step synthesis starting with cyclopentylbromide **290** (Scheme 52) [172]. Nucleophilic substitution of the bromine atom affords a nitro compound with a moderate yield. Remarkably, the use of iodine as an oxidant in the dimerization reaction of **291** causes a significant increase in the yield of dinitroalkane **292** (from 17% to 50%) as compared to a previously described procedure involving Pb(OAc)_4_ [173]. The reduction of vicinal dinitrocompound **292** by the well-known Zn/NH_4_Cl system and subsequent condensation of bis(hydroxylamine) sulphate **293** with heteroaromatic aldehydes, followed by the oxidation of crude *N,N**′*-dihydroxyimidazolidine **294** with manganese dioxide or sodium periodate, afford di-SNNRs **295a–d**. Based on the pyrazolyl-substituted SNR **295d**, its *N*-ethyl derivative **296d** is obtained. The resultant radicals **295a–c** and **296d** have been employed for the synthesis of various heterospin complexes with unusual spin transitions that are observed when temperature or pressure changes [174,175,176].

One of the rare examples of the synthesis of an iminonitroxyl radical with a spiro junction at the β-carbon atom is demonstrated in reference [177]. Namely, condensation of 3-hydroxyamino-3-methylbutan-2-one with benzaldehyde in the presence of ammonia produces 1-hydroxy-2,5-dihydroimidazole **297**, followed by metalation and subsequent interaction with ethyl benzoate, leading to enaminoketone **298**. Brief oxidation of the latter with manganese dioxide (otherwise, further oxidation of the methylene moiety occurs, with the formation of dimer **300**) results in 4*H*-imidazole 3-oxide **299** in a quantitative yield. Oximation of the latter and a reaction of the intermediate with MnO_2_ respectively cause the closure of the isoxazole ring and the oxidation of cyclic hydroxylamine to SNR **301** (Scheme 53).

## 8. Imidazolidine-Type SNRs

To obtain promising pH-sensitive spin labels, Keana et al. developed a simple method for the synthesis of SNRs of the imidazolidine series having structural formulas **302** and **303** (Figure 10) [25,26,27].

Condensation of 1,2-diamino-2-methylpropane with cyclohexanone or *N*-protected piperidones **58a,d,g** in the presence of PTSA generates corresponding imidazolidines **304a,d,g** in a high yield. Their immediate protection by formylation with a mixed anhydride on the less sterically hindered nitrogen and the oxidation of intermediate formyl derivatives **305a,d,g** with *m*-CPBA lead to SNRs **306a,d,g** containing a formamide group. Their subsequent deprotection by hydrolysis in an aqeous–methanolic solution of KOH generates desired nitroxides **302a,b** quantitatively (Scheme 54) [26,27].

For the synthesis of SNRs of type **303**, ketones **58a,d** are condensed with 2,3-diamino-2,3-dimethylbutane, and the resulting imidazolidines **307a,d** are oxidized with *m*-CPBA to corresponding nitroxides **303a,d** without the use of any protective groups. Alkaline hydrolysis of the acetyl residue in SNR **303d** produces diamino radical **303b** (Scheme 55) [27].

Using nitroxide **303a**, Keana et al. carried out a further sequence of transformations, which make it possible to obtain unique diradical **308** [25], whose structure has been confirmed by X-ray diffraction analysis (Figure 11). To this end, nitroxide **303a** is reduced in the H_2_/Pd/C system to the corresponding hydroxylamine derivative, the treatment of which with AcCl/Et_3_N without isolation from the reaction mixture provides acyl derivative **309.** Radical **310** is obtained in a high yield by the oxidation of amine **309**. Careful treatment of **310** with a methanolic alkali gives *N*-hydroxyradical **311** in a 56% yield. Final oxidation of intermediate **311** is conducted by the bubbling of oxygen gas for 3 min through a solution of the radical in the *^t^*BuOH–*^t^*BuOK system, making it possible to obtain diradical **308** in a high yield. A specific feature of this diradical is the presence of two unpaired electrons on geminal nitroxyl groups entailing a strong exchange interaction.

A similar approach was employed to synthesize a related SNDR based on a cyclic ketone, 5α-cholestan-3-one [25].

Another technique for the synthesis of imidazolidine SNRs, proposed by Reznikov, involves the addition of organolithium compounds to cyclic nitrones of the 2,5-dihydroimidazole series and allows for the introduction of substituents other than the methyl group at the C-4 atom of the imidazolidine ring. Thus, by alkylation of 5,5-dimethyl-2-spirocyclohexyl-4-phenyl- 3-imidazoline-3-oxide **283 [167]** with a mixture of formic acid and formaldehyde, *N*-methyl derivative **312** is obtained, whose subsequent reaction with PhLi and oxidation of intermediate hydroxylamine **313** (without isolation) by MnO_2_ lead to nitroxide **314** with a quantitative yield (Scheme 56) [178]. On the other hand, to synthesize spirocyclic diradical **315**, which cannot be obtained by direct oxidation of mono-SNR **314**, a different starting substrate **280** is utilized. Treatment with phenyllithium results in a crystalline *N,N′*-dihydroxy derivative **316**, which is oxidized by the original technique of slurrying in pentane and aging over manganese dioxide for a short period to prevent overoxidation. In contrast to tetramethyl analogs [178], evaporation or several-minute ambient-temperature incubation of a solution of diradical **315** causes its complete decomposition, which is accompanied by a release of nitrogen oxides and benzophenon formation. Apparently, the presence of two phenyl substituents and a spirocycle, which are larger than the methyl groups, creates serious steric hindrances that significantly reduce the stability of diradical **315**.

Peculiar SNRs of the imidazolidine series were obtained from 1,2,2,4,5,5-hexamethyl-3-imidazoline 3-oxide **317** in a study on the possibility of CH_3_ group functionalization at the C-4 position of the heterocycle. Metallation of methylnitrone **317** with PhLi, followed by treatment with an arylcarboxylic acid ester, produced β-oxonitrone **318**. A reaction of the latter with hydroxylamine yields oxime **319**, which exists in solution in equilibrium with a spirocyclic tautomeric form, compound **320**. It is not surprising that the oxidation of compound **320** by manganese dioxide gives rise to stable SNR **321** (Scheme 57). To establish the limits of applicability of the observed oxidative cyclization, reactions of nitrone **318** with other nitrogenous nucleophiles (hydrazine, semicarbazide, and thiosemicarbazide) have been investigated. It was revealed that only in the case of semicarbazide can the resulting intermediate **322** be oxidized to a stable SNR, whereas the other derivatives undergo imidazolidine ring opening under oxidation conditions. The structure of the obtained nitroxide **323** has been proved by an analysis of its EPR spectrum, which is complicated due to splitting at three nonequivalent nitrogen atoms (Figure 12), which constitutes evidence for the formation of a spiro node with the nitrogen atom attached to the α-carbon atom of the nitroxyl group [179].

Sequential alkylation of nitroxides of the 3-imidazoline series **324a–d** at the *sp^2^*-nitrogen atom and subsequent reduction of intermediate imidazolinium salts are another way to synthesize SNRs of the imidazolidine series. For instance, methylation of **324a–d** with dimethyl sulfate and treatment of the resulting iminium salts **325a–d** with sodium borohydride lead to **SNRs 326a–d** in 30–90% yields [155,180]. On the other hand, if the obtained iminium salt **325d–f** is treated with a base, then the corresponding enamine forms, which can condense with salicylic aldehyde or 2-hydroxynaphthaldehyde, lead—due to intramolecular nucleophilic cyclization—to paramagnetic spiropyrans **327d–f** (Scheme 58), which are representatives of β-SNRs and promising spin probes for biophysical studies [181].

One of the common methods for the synthesis of functionalized imidazolidine radicals containing a keto group near the amine nitrogen atom is the cyclization reaction of α-aminonitriles **111, 328** with carbonyl compounds under basic catalysis conditions, followed by oxidation of the resulting amines **329** to SNR. Thus, in particular, a large series of SNRs of the imidazolidinone series was obtained—**330a–t** (Scheme 59, Table 2) [182]. The resultant radicals are very stable, and no change occurred after storage at ambient temperature for several years. In the case of cyclic amines **329b,e,f,j–l,n–p,r,** which contain an α-hydrogen adjacent to the secondary amino group, oxidation products are nonradicals. Alkyl derivatives of SNRs **331a,b** have also been synthesized to be applied as mediators in styrene radical polymerization [183].

According to the proposed mechanism behind the cyclization reaction of α-aminonitriles with carbonyl compounds, the amino group of substrate **111** initially attacks the carbonyl carbon atom of the ketone, with the subsequent addition of a hydroxy anion to the cyano group, thus triggering cyclization with the formation of imidazolidinone **329a** (Scheme 60) [184].

A slightly different approach to the synthesis of SNDR **332** of the imidazolidinone series was used in reference [77] (Scheme 61). Aminonitrile **328** (R_1 + 2_ = 4,4-TMP) was hydrolyzed in the presence of sulfuric acid to aminoamide **333**, and subsequent acid-catalyzed cyclization of the latter in a mixture of acetone and its dimethyl ketal [185] led quantitatively to bicycle **334**. Double oxidation of diamine **334** with peracetic acid resulted in diradical **332**, which was employed as a monomer for constructing an electroactive paramagnetic polymer to create a rechargeable organic radical battery [77].

Finally, an original approach to paramagnetic *N*-methylimidazolidinones was devised by French researchers on the basis of an ether of the simplest amino acid, glycine. Its application to a three-step process consisting of amidation, condensation, and oxidation of cyclic amine **335** resulted in aldonitrone **336** [186]. Pd-catalyzed addition of arylbromide to cyclic nitrone **336** in the presence of pivalic acid provided ketonitrone **337**, and subsequent Grignard treatment and oxidation of the intermediate hydroxylamine afforded SNR **338** (Scheme 62) [187].

## 9. Oxazolidine (DOXYL) SNRs

The main technique for the synthesis of oxazolidine nitroxyl radicals **339** (DOXYLs) is the approach developed by Keana et al. [30], which involves the condensation of a readily available 1,2-amino alcohol (2-amino-2-methylpropan-1-ol **340**) with dialkyl ketones, followed by oxidation of the secondary amino group in cyclic adduct **341** with organic or inorganic peroxides (Scheme 63).

The proposed method has been utilized to obtain spirocyclic nitroxides **342**, **345 [30]**; **343**, **344 [188]**, **346 [31]**, and **347 [189]**, derivatives of cyclohexanone and 3-keto steroids (Figure 13).

Spiro-fused heterodiradical **348** is obtained by the condensation of 2,2,6,6-tetramethyl-4-piperidone **6** with amino alcohol **340** followed by simultaneous oxidation of amino groups in five- and six-membered heterocycles of **349** with the formation of α,γ-SNDR (Scheme 64, Figure 14) [190].

Although the synthesis of symmetric SNDRs with a rigid cyclic linker of the piperidine, morpholine, pyrrolidine, and imidazoline series requires the use of an iterative, often a multistep pathway (see Scheme 13, Scheme 29, Scheme 38 and Scheme 45), the preparation of a DOXYL-type SNDR is a simple procedure, consisting of the condensation of 2 moles of an amino alcohol with 1,4-diketone with subsequent oxidation of the intermediate diamine. For example, refluxing 2-amino-2-methylpropan-1-ol **340** with 1,4-cyclohexanedione **157a** in the presence of PTSA readily leads to cyclic *trans*-bisamine **350**, the treatment of which with *m*-CPBA generates the *trans*-isomer of SNDR **351** in a 50% yield [191]. When the methyltrioxorhenium/H_2_O_2_ system is employed at the second step as an oxidizer, the diradical yield can be increased to 80%, [106], and in an Oxone/acetone mixture, this yield can reach 90% [105] (Scheme 65). An X-ray study of SNDR **351** revealed that both *N*-oxyl groups in the molecule are *trans*-diequatorial, with an intramolecular distance for the N atoms of 5.75 Å and the oxygens of 7.00 Å [192].

A number of di-SNR DOXYLs **361–369** have been obtained in a similar manner when the above synthetic scheme has been applied to isomeric decalinediones **352–354**, bis(4-cyclohexanone) **355**, spiro[5.5]undecane-3,9-dione **356**, and steroid diketones **357–360** (Table 3).

For trispirocyclic bisnitroxide **365**, which is obtained as a polarization agent for DNP [43], the relative mutual arrangement of NO groups in the biradical has been confirmed by X-ray diffraction (Figure 15) [193]. Due to the spiro junctions joining the rings, compound **365** has a rigid structure, and the odd number of spiro centers forces the nitroxide moieties to be almost orthogonal (*θ*, the angle between the mean planes of DOXYL moieties, is 88°).

To determine whether the major contribution to the exchange in rigid diradicals is mediated by space (via a direct orbital overlap) or through the multi-sigma-bond pathway between the paramagnetic subunits, SNR DOXYL-steroid nitroxide biradical **370** has been synthesized based on a steroid molecule of isoandrololactam acetate **371** (Scheme 66). Accordingly, treatment of the latter with dimethyl sulfate gave *O*-methyl ether **372** and the subsequent Grignard addition to allyl magnesium bromide afforded cyclic amino alcohol **373** with a low yield [196]. Using a modified Sarret procedure (a complex of CrO_3_ prepared in situ with pyridine in DCM), a secondary hydroxy group was converted into a keto group and then ketoamine **374** was transformed into biradical **370** via a standard sequence, i.e., condensation with **340**, followed by simultaneous oxidation of both amino groups in **375** to nitroxide moieties [197].

Highly hydrophilic di-SNR heterodiradical **Isodoxa 376** (constructed from different types of nitroxides, an **Isoindoline** and **DOXYL** nucleus, linked to each other via a rigid spirocyclic linker) was synthesized as a useful paramagnetic unit to study the formation of supramolecular polyradical structures based on triangular assemblies involving cucurbit[8]uril.

This diradical was obtained by convergent synthesis involving the reaction of a suitable functionalized SNR of the DOXYL type **377** [27] with protected isoindolinoxyl **378** before the cleavage of the protective acetyl group (Scheme 67). **Isodoxa 376** is characterized by an EPR spectrum with 15 lines, and its molecular structure has been resolved by X-ray analysis (CCDC 1872438) [97]. 

In addition to 2-amino-2-methylpropan-1-ol **340** being used for the synthesis of SNRs of the DOXYL type, other different 1,2-aminoalcohols prepared from simple or complex natural compounds can also participate in spirocyclization reactions. Below, we will consider examples of such syntheses.

Amino alcohol **379** based on 2-adamantanone was synthesized in several steps through intermediate 2-nitroderivative **194** [198]. Scheme 68 depicts one of the most optimal versions of its synthesis [199].

Amino alcohol **379** is condensed in an autoclave with different ketones to obtain sterically hindered mono- and dispirooxazolidines, with subsequent oxidation with *m*-CPBA to obtain target Ad-conjugated SNRs **381** and **384** with low to moderate yields (Scheme 69) [200]. It should be noted that, in the case of oxazolidine **382**, no corresponding SNR was obtained, possibly due to steric hindrances in the diamagnetic precursor.

SNR **385** was synthesized for use as a catalyst in living polymerization processes. Namely, 1-amino-1-cyclohexanecarboxylic acid **386** was reduced with an excess of LAH to the corresponding amino alcohol **387**, followed by a reaction with cyclohexanone in the presence of TsOH, quantitatively affording amine **388** (Scheme 70). Oxidation of the latter with *m*-CPBA led to nitroxide **385** in a low yield [66].

The amino acid **390** required for the synthesis of 1,2-aminoalcohol can also be obtained from hydantoin, an adduct of the Bucherer–Bergs reaction [32]. In this way, on the basis of 4-piperidone **6** through the formation of spirocyclic intermediate **389**, an amino alcohol, (4-amino-2,2,6,6-tetramethylpiperidin-4-yl)methanol **391**, was obtained (Scheme 71). The condensation of the latter with cyclic ketones **a–c** and acetone **d** leads with high yields to diamines **392a–d**; their subsequent oxidation with two equivalent of *m*-CPBA produces mono-SNR diradical **393d** [43] and di-SNR diradicals **393a** [201,202], **393b** [203], and **393c [32]**.

Via the above approach, steroid-type nitroxide **394** was obtained from 5α-cholestan-3-one **395** as a mixture of stereoisomers in a ~7:1 ratio. Isomer **394a**, in which the NO bond is at the equatorial position of the cyclohexane ring, is predominant [33] (Scheme 72).

Spirocyclic oxazolidinones can also serve as sources of 1,2-amino alcohols. For instance, based on camphene, chiral SNRs camphoxyls, i.e., DOXYLs, spiro-conjugated with a natural compound were obtained [204,205]. Scheme 73 presents four-step synthesis of (1*R*,2*S*,4*S*)-3,3-dimethylspiro[bicyclo[2.2.1]heptane-2,4′-oxazolidin]-2′-one (+)-**396** [206]. (−)-Camphene is converted via hydride reduction, phosgenation, azide formation, and solvent thermolysis to an endo:exo mixture of oxazolidinones (*+*)-**396** and (*+*)-**397**, from which (+)-**396** is isolated by fractional crystallization. Alkali-promoted hydrolysis of **396** quantitatively affords corresponding amino alcohol **398**. Due to the reduced activity of the amino group in **398**, which is explained by the localization of the nitrogen atom at the neopentyl position, acid-catalyzed acetalization for forming the oxazolidine **399** by means of acetone fails but proceeds smoothly with 2,2-dimethoxypropane, thereby giving cyclic amine **399** in a 70% yield. Its subsequent oxidation with *m*-CPBA leads to camphoxyl radical (*+*)-**400**, which is isolated in a moderate yield as an orange crystalline compound, stable for at least one year when stored in a freezer. Similarly, from commercially available (−)-**396**, SNR (*−*)-**400** was prepared [206]. Of note, in the case of benzophenone acetal, although cyclic amine **401** is formed in a low yield; it fails to be oxidized to the corresponding radical **402** either by *m*-CPBA or by dimethyldioxirane. To alleviate this problem, geminal isobutyl camphoxyl derivative **403** was prepared (Scheme 73). In this case, oxidation of **404** with *m*-CPBA proceeded in a 65% yield to provide **403** as a stable crystalline solid [205].

Another method for the synthesis of oxazolidine SNRs is the oxidation of cyclic hydroxylamines with manganese dioxide. These hydroxylamines are usually formed by the condensation of 2-hydroxylamino alcohols with ketones in the presence of ammonium acetate. The drawbacks of this method include lower availability of 2-hydroxylamino alcohols compared to 2-aminoalcohols; however, the advantages of this approach are a much shorter time of condensation with carbonyl compounds and the ease of oxidation of hydroxylamine derivatives as compared to sterically hindered amines. For example, the condensation of 2-(hydroxyamino)-2-methyl-1- phenylpropan-1-ol **405** with cyclohexanone is completed within 2 h, affording oxazolidine **406** in an 80% yield. Oxidation of the latter with MnO_2_ or air oxygen produces target radical **407** in a quantitative yield (Scheme 74) [166].

It is noteworthy that the catalytic activity of ammonium acetate has also been demonstrated for the condensation of 2-amino-2-methylpropanol **340** with cyclic ketones. It is reported that refluxing of a solution of 5α-cholestan-3-one **395** with a three-fold excess of amino alcohol **340** in the presence of one equivalent of NH_4_OAc for 2 h results in the corresponding oxazolidine in a 70% yield [166].

## 10. SNRs of Other Types

In the literature, there are few reports of the synthesis of SNRs where the heterocyclic paramagnetic frame belongs to classes not described above; however, these examples, including both classic and nontraditional approaches to the synthesis of nitroxides, may prove to be useful and interesting to the reader; therefore, they are considered in a separate section.

Stable indolinone-type nitroxide radical **408** is obtained via oxidation of 1-hydroxy-2′,6′-diphenylspiro[indoline-2,4′-pyran]-3-one **409** [207]. The latter is prepared as a product of intramolecular redox photocyclization of 2,6-diphenyl-4-(*o*-nitrobenzylidene)-4*H*-pyran **410**. In turn, such a pyran is easily available from a reaction of 4-methoxy–substituted pyrylium salt **411** with *o*-nitrophenylacetic acid **412** in the presence of a tertiary amine (Scheme 75) [208].

A similar SNR of the indolinone type with a quinone methide fragment, radical **413**, is synthesized through oxidation of parent cyclic hydroxylamine **414**, which in turn is obtained via unusual reductive cyclization of a tetrasubstituted benzophenone **415**, containing an *ortho*-nitro group on one aromatic ring and two methoxy groups and a phenolic moiety on the other [209]. The authors of that study suggested that the most promising way to close the five-membered heterocycle is to first form intermediate nitroso derivative **416**, which is captured by nucleophilic addition of the electron-rich aryl ring to the nitroso group before it can be reduced further to the corresponding hydroxylamine (Scheme 76). This case represents a rare example of reductive phenolic coupling in contrast to numerous examples of oxidative phenolic coupling. The correctness of the structural assignment of cyclic hydroxylamine **414** has been confirmed by X-ray crystallographic analysis. Moreover, the characteristics of an ESR spectrum of SNR **413** in some details resemble those of radical **408**. For instance, the experimental hyperfine coupling constants that were determined for nitrogen and aromatic hydrogens in **413** and **408** are *a*_N_ = 0.965 mT and 0.930 mT; *a*_H4,7_ = 0.316 mT and 0.310 mT; and *a*_H5,6_ = 0.104 mT and 0.100 mT, respectively. Furthermore, an unusual feature of the **413** spectrum is a small long-range splitting of 0.06 mT, which is due to the two dienone protons.

Benzo[*d*][1,3]oxazine-type SNR **417** was obtained via three-stage synthesis from anthranilic acid [210] (Scheme 77). Nitroxide **417**, although it was isolated as a crystalline red compound with melting point 88 °C, proved to be unstable at room temperature and could be preserved without decomposition only at a liquid-nitrogen temperature. A possible reason for such liability is the presence of reactive hydrogen atoms at activated *para*- and benzyl positions (6-H and 8-H).

Triplet (S = 1) stable diradical **418** comes from the same benzoxazine family but is devoid of the shortcomings of SNR **417**; **417** was synthesized relatively recently [80]. Scheme 78 illustrates a multistep route of its synthesis. In this way, dimethyl 4,6-dibromoisophtalate **419**, which is prepared from 4,6-dibromo-*meta*-xylene, is introduced into Pd-catalyzed C–N cross-coupling with an excess of benzylamine.

Subsequent reductive debenzylation of compound **420** to diamine **421** and a reaction of the latter with a large excess of MeMgBr allow us to obtain diaminodiol **422** quantitatively. Of note, double condensation of **422** with cyclohexanone in the presence of AcOH leads to cyclic diamine **423** in a 34% yield, whereas the catalysis by silica gel in the absence of an organic acid and without heating increases the yield of target product **423** at this stage to 78% [211]. By oxidation of **423** with *m*-CPBA, SNDR **418** is obtained with a yield of 30%. 1,9-Diaza-3,7-dioxaanthracene-1,9-dioxyl **418** and other diradicals of this series possess robust triplet ground states with strong ferromagnetic coupling and good stability under ambient conditions.

A curious SNR of the tetrahydropyrazine series, radical **424**, was synthesized in a study on the reactivity of sterically encumbered 2,2,3,5,5,6-hexamethyl-2,5-dihydropyrazine *N,N′*-dioxide **249** (R_1–3_ = Me) [212]. Accordingly, when compound **249** is treated with PhLi, preferential metalation of only one of the activated methyl groups takes place; a subsequent reaction of the intermediate with ethyl benzoate causes the formation of a monoacylation product, ketone **425**, with a yield of 30%. The reaction of the latter with hydroxylamine generates oxime **426**, which, as in the case of pyrroline 1-oxide **244** (Scheme 42), exists in solution mainly in the form of the spirocyclic tautomeric form **427**. Oxidation of spirocyclic hydroxylamine **427** with MnO_2_ quantitatively affords SNR **424** (Scheme 79).

The preparation of nitroxides capable of integration into special structures, such as fullerenes, is of interest for the design of organic ferromagnetic materials. Chinese authors found that cyclic ketones, like aldehydes, are capable of reacting with 2-aminoisobutyric acid when boiled in chlorobenzene thereby forming reactive azomethine ylide, which then reacts with the double bond of C_60_ (C_70_) in 1,3-dipolar cycloaddition to yield the final [60]-fulleropyrrolidine **428** or [70]-fulleropyrrolidine **429** [213]. Corresponding stable nitroxides annelated with fullerenes (C_60_, C_70_), **430** and **431**, are obtained via the oxidation of amine derivatives **428** and **429** by an excess of *m*-CPBA (Scheme 80) [214].

## 11. Conclusions

In this review, we tried to cover diverse approaches to the preparation of SNRs of various five- and six-membered nitrogen heterocycles, ranging from the first historical experiments up to the most recent advances involving complex transformations based on organometallic species and stereoselective synthetic methods. The preparation of SNRs containing at least one nitrogen atom and 0–2 oxygen atoms in the following heterocyclic nuclei was reviewed here; piperidine, tetrahydroquinoline, spirobiacridine, piperazine, morpholine, pyrroline, pyrrolidine, 2- and 3-imidazoline, imidazolidine, oxazolidine, and sporadic examples of the synthesis of other SNR types.

As a rule, the synthesis of SNRs consists mainly of choosing the proper method for obtaining a diamagnetic precursor because the final oxidative stage poses no special difficulties. In the synthesis of spirocyclic amines (hydroxylamines), there are several main approaches, such as (a) recyclizations of heterocycles; (b) condensations of bifunctional nitrogen-containing nucleophiles with carbonyl compounds, whereas for the synthesis of bis-nitroxides, a step-by-step strategy of molecule assembly is often chosen; and (c) the formation of a spiro junction in a multistep procedure—aldonitrone → ketonitrone with a terminal alkene function → 1,3-dipolar intramolecular cycloaddition → N-O bond breakage in the resulting adduct. Other methods of designing the SNR spiro framework, such as the Diels–Alder reaction of imines, acid-catalyzed reactions of aminonitriles, oxidative alkoxylation of nitrones, and a shift of the tautomeric equilibrium of oxime-nitrone/*N*-hydroxy-isoxazoline toward the ring at the oxidation step, are quite rare in the practice of SNR synthesis.

Most of the SNRs mentioned in the review were synthesized in original studies with the aim of their subsequent application in various fields of natural sciences. An upcoming review of uses of the most diverse SNRs, including functionalized ones, is expected to clarify the current situation and the place of spirocyclic nitroxides in the chemistry of materials and various biological and biochemical applications.

## Data Availability

The data presented in this study are available on request from the corresponding author.

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
