# Peer review of "Spirocyclic Nitroxides as Versatile Tools in Modern Natural Sciences: From Synthesis to Applications. Part I. Old and New Synthetic Approaches to Spirocyclic Nitroxyl Radicals"

_molecules, 2021, doi:10.3390/molecules26030677_

Round 1

Reviewer 1 Report

Nitroxyl radicals are one of important spin source for stable organic radicals, and have been widely used in many fields. In this review article, the authors focused on the synthetic aspects with a brief explanation of applications of cyclic nitroxyl radials bearing spirocyclic moiety. They classified these compounds into several categories based on the structure of precursor amines such as piperidine, tetrahydroquinoline, piperazine, morpholine, pyrroline, imidazoline, oxazolidine, and so on. This is somewhat “nitche” subject. However, the manuscript is well-organized and would be useful for researchers who are interested in the related molecules. Thus, I can recommend the publications after addressing and correcting the following minor points.

-The abbreviation of DNP is used not only for dynamic nuclear polarization but for dialkoxynaphthalene (p.6). This is confusing and it should not be used for the latter.

-Both “diradical” and “biradical are not distinguished in this manuscript. In fact, many researchers used these terms as synonym, but IUPAC explains these are different. See IUPAC GOLD book  https://goldbook.iupac.org/terms/view/B00671. A brief explanation is necessary in this manuscript. Or unify the terms.

-In l. 147, 30a-c??

-In l. 779, one ore?? Two

-The definition of compound 328 in Scheme 59 appears to be vague. Suddenly, 328a is shown in Scheme 61. Then, which compounds are used for the results shown in Table 2?

-In l. 1111, xx and zz??

Reviewer 2 Report

This review by Elena V. Zaytseva and Dmitrii G. Mazhukin is dealing with diverse approaches to the preparation of SNRs of various 5- and 6-membered nitrogen heterocycles, ranging from the first historical experiments up to the most recent  advances involving complex transformations based on organometallic species and stereoselective synthetic methods. The upcoming review of uses and application of the diverse SNRs, I believe it will be very interesting.

In my opinion it is very complete and fully up-to-date, efficiently detailing the synthetic routes of SNRs and most of the mechanisms involved. In my opinion, it can be published once the indications made are corrected, some of which are important even when accepted after minor review.

This reviewer considers that this review is exactly worthy of publication in Molecules after below points:

1) The review is very extensive and it would be advisable to start with an index indicating the 10 sections ang pages where they are discussed and, where appropriate, the different subsections to facilitate revisiting the text.

2) The list of abbreviations must be complete with all those that appear in the text, for example:

  • DEER (pg 2, line 53)
  • TEMPOL (pg 5, line 162)
  • TEKPOL (pg 19, line 539)
  • ORCA (pg 22, line 603), KN (line611)
  • HAK (pg 28, line 761)
  •  

3) Scheme 2 should include the mechanism of formation of dispriocyclic 8, and in step 13 to 7 the elimination of ketone that explains the formation of 6 should be indicated.

4) - Page 5, line 147 intead of 30a-c, should say 23a-c.

    - Page 34, line 903 intead of 2-hydroxyamino-2-methylbutan-3-one, should say 3-hydroxyamino-3-methylbutan-2-one.

    - Page 35, line 932 intead of 278, should say 308.

     - Page 40, line 1058 and Scheme 65, cyclic bisamine 350 should be change to cyclic trans-bisamine 350.

     - Page 38, Table 2, in the foot should say ND: yield not determined.

5) The formulas throughout the text are represented with the diminished numbers and in the schemes with a subscript, in my opinion it is the journal's policy to homogenize them as subscripts.

Ej. pg 5, line 157: EDTA-Na2 and Na2WO4 and in Scheme 4, EDTA-Na2 and Na2WO4

and this applies to the entire article.

6) - Pg 9, line 254, for Grob-type fragmentation a reference is requiered.

    - Pg 18, line 516, for Bucherer–Bergs reaction a reference is requiered.

    - Pg 43, line 1099, probably a reference is requiered for X-ray analysis (CCDC 1872438).

7) Scheme 11 must be completed including the elimination of the by-product that produces 2 molecules of ketone and methylamine, in step 68 to 69.

8) Pg 12, line 347, SNDR 88, this compound is not shown in schemes.

9) I don't understand the sentence on page 22, line 615-617, could the author rephrase it?:

"Sequential hydrolysis of the latter either in the form of a mixture of isomers or as each isomer separately yields a single product: trans-dicarboxylic acid 182".

In the same way the following phrase sounds strange (pg 26, line 720):

"Funny that the second regioisomer, aldonitrone 226, is not a byproduct in this scheme"

10) In scheme 34 (pg 23) it should be discussed whether another regioisomer different from 191 is obtained in the cycloaddition reaction and why this is obtained (is it steric interaction?), Since the organometallic nucleophiles attack the carbon of the dipole.

11) In Scheme 41, pg 28, the compound from 240 to 239 is an intermediate, which has not been isolated and must be in square brackets.

12) In Scheme 45, pg 30, the ketone 1,4-cyclohexandione 157a (it does not appear from page 20) can be included indicating how it can lead to 58f and the direct reaction does not allow to obtain 260.

13) In Scheme 59, p. 38, compounds 329a-t are incorrect, the isomer where NH and CO are changed should be shown, according to the mechanism shown in Scheme 60. And in this Scheme 60, the arrow coming from the nitrile must be corrected close to the N, here it appears that the C is forming the new bond.

14) In Fig 13, pg 40, the blue data for 342 is missing.

15) The cyclization from 415 to 414 is very interesting and unusual (pg 46, Scheme 76), I would appreciate, if possible, a further explanation of the mechanism and if it can be the NO2 group itself (in 415) activated by Zn that participates in the cyclization.
